# An In-Depth Study on the Chemical Composition and Biological Effects of *Pelargonium graveolens* Essential Oil

**DOI:** 10.3390/foods13010033

**Published:** 2023-12-21

**Authors:** Miroslava Kačániová, Milena Vukic, Nenad L. Vukovic, Natália Čmiková, Andrea Verešová, Marianna Schwarzová, Mária Babošová, Jana Ivanič Porhajašová, Maciej Kluz, Bożena Waszkiewicz-Robak, Anis Ben Hsouna, Rania Ben Saad, Stefania Garzoli

**Affiliations:** 1Institute of Horticulture, Faculty of Horticulture and Landscape Engineering, Slovak University of Agriculture, Tr. A. Hlinku 2, 949 76 Nitra, Slovakia; 2School of Medical & Health Sciences, University of Economics and Human Sciences in Warsaw, Okopowa 59, 01 043 Warszawa, Poland; 3Department of Chemistry, Faculty of Science, University of Kragujevac, 34000 Kragujevac, Serbia; 4Institute of Plant and Environmental Sciences, Faculty of Agrobiology and Food Resources, Slovak University of Agriculture, Tr. A. Hlinku 2, 949 76 Nitra, Slovakia; 5Laboratory of Biotechnology and Plant Improvement, Centre of Biotechnology of Sfax, B.P “1177”, Sfax 3018, Tunisia; benhsounanis@gmail.com (A.B.H.); raniabensaad@gmail.com (R.B.S.); 6Department of Environmental Sciences and Nutrition, Higher Institute of Applied Sciences and Technology of Mahdia, University of Monastir, Monastir 5000, Tunisia; 7Department of Chemistry and Technologies of Drug, Sapienza University, P. le Aldo Moro, 5, 00185 Rome, Italy; stefania.garzoli@uniroma1.it

**Keywords:** geranium essential oil, chemical composition, antimicrobial activity, vapor phase, mass spectrometry, *Harmonia axyridis*

## Abstract

The essential oil of *Pelargonium graveolens* (PGEO) is identified in the literature as a rich source of bioactive compounds with a high level of biological activity. This study aimed to examine the chemical profile of PGEO as well as its antioxidant, antibacterial, antibiofilm, and insecticidal properties. Its chemical composition was analyzed using gas chromatography–mass spectrometry (GC-MS), achieving comprehensive identification of 99.2% of volatile compounds. The predominant identified compounds were β-citronellol (29.7%) and geraniol (14.6%). PGEO’s antioxidant potential was determined by means of DPPH radical and ABTS radical cation neutralization. The results indicate a higher capacity of PGEO to neutralize the ABTS radical cation, with an IC_50_ value of 0.26 ± 0.02 mg/mL. Two techniques were used to assess antimicrobial activity: minimum inhibitory concentration (MIC) and disk diffusion. Antimicrobial evaluation using the disk diffusion method revealed that *Salmonella enterica* (14.33 ± 0.58 mm), which forms biofilms, and *Priestia megaterium* (14.67 ± 0.58 mm) were most susceptible to exposure to PGEO. The MIC assay demonstrated the highest performance of this EO against biofilm-forming *S. enterica* (MIC 50 0.57 ± 0.006; MIC 90 0.169 ± 0.08 mg/mL). In contrast to contact application, the assessment of the *in situ* vapor phase antibacterial activity of PGEO revealed significantly more potent effects. An analysis of antibiofilm activity using MALDI-TOF MS demonstrated PGEO’s capacity to disrupt the biofilm homeostasis of *S. enterica* growing on plastic and stainless steel. Additionally, insecticidal evaluations indicated that treatment with PGEO at doses of 100% and 50% resulted in the complete mortality of all *Harmonia axyridis* individuals.

## 1. Introduction

For centuries, the aromatic and therapeutic properties of plants and their extracts have been harnessed. In recent times there has been growing interest in bioactive agents, with particular attention paid to compounds of natural origin. Essential oils (EOs), recognized as odoriferous or volatile oils, have garnered significant attention due to their antimicrobial and antioxidant effects. This heightened interest is largely attributed to global advancements and the rise of various illnesses associated with modern civilization [1,2,3,4,5,6]. They represent highly complicated mixtures, often comprising a few to several hundred individual compounds. These are volatile, fragrant, oily liquids derived from various plant parts, mostly non-woody ones like flowers, leaves, peels, buds, and seeds, but also woody sections like bark or roots, where the glands and ducts that contain the EOs are found [2,3,4,5,6,7]. Volatile oils have wide application in many industries. They are incorporated as natural preservatives or perfumes in most cosmetic products. Also, they are employed as natural additives in foods and food packaging materials with the goal of protecting them from oxidative damage and inflammatory diseases. Furthermore, due to their purported antiviral, nematicidal, antifungal, and insecticidal qualities, EOs have replaced synthetic materials in the domains of agriculture, pharmaceuticals, and nutrition [2,4,7,8,9].

*Pelargonium graveolens* is a fragrant, hairy shrub that grows upright to a height of 1.3 m and can spread to 1 m. Its leaves are carved and soft, and it typically produces tiny, pink flowers. *Pelargonium* species are frequently employed in the biosynthesis of EOs due to their unique qualities and variety of odors. Their production of EOs is substantial, ranging from 0.1% to 0.9% *v*/*w* [10]. In most cases, the leaves, petals, and stalks are used to extract the oil [11]. Research indicates that the composition of rose-scented geranium is affected by a number of variables, including the cultivar, the process of distilling the oil, the portion of the plant that has been distilled, the age of the material, the storage of the EOs, the place of growth of the plant and seasonal variations in the area (temperature, light intensity), as well as the season and time of harvest [12].

Chemical control is primarily used to manage pathogenic bacteria and food spoilage. However, its use is limited due to unfavorable characteristics, such as carcinogenicity, acute toxicity, teratogenicity, and slow degradation times, which may cause pollution and other environmental issues [13]. Moreover, unfavorable public opinion on commercially used food-grade antimicrobials has sparked additional interest in using more naturally occurring substances [14]. Presently, extensive research is underway to identify potential candidates among natural food additives capable of prolonging the shelf life and improving the quality of perishable foods, all while preserving a broad spectrum of antioxidant and antibacterial properties [15]. Within this realm, there is burgeoning interest in the utilization of EOs as alternative agents to regulate food spoilage and combat harmful pathogens [16,17,18].

Moreover, the formation of microbial biofilms, capable of disrupting industrial processes, poses a challenge for the food processing sector. The protective mechanisms employed by microbial cells within biofilms are intricate and differ from those observed in planktonic cells. These mechanisms encompass an elevated horizontal transfer rate of resistance genes, matrix impermeability, an altered transcription rate, persistent cell selection, the accumulation of antibiotic-inactivating enzymes, and other complex adaptations [19]. As a result, compared to planktonic cells, biofilm cells can develop up to 10–1000 times greater resistance to antimicrobial agents [20]. The risk of contamination with pathogenic bacteria (such as *Salmonella enterica*, *Listeria monocytogenes*, and *Escherichia coli*), which mainly grow in biofilms, makes food-contact surfaces one of the main sources of problems for the industry. The problem of disinfecting surfaces that come into contact with food is complex and demanding, but can be resolved by developing new disinfectants [21,22,23].

Regarding everything mentioned above, the main goals of this work were to determine the chemical composition of *Pelargonium graveolens* essential oil (PGEO) using GC-MS analysis, and examine its antioxidant, antibacterial (*in vitro* and *in situ*), antibiofilm, and insecticidal activity.

## 2. Materials and Methods

### 2.1. Pelargonium graveolens Essential Oil

The EO utilized in this study was extracted through the steam distillation of fresh wort from *Pelargonium graveolens* and was procured from Hanus s.r.o. (Nitra, Slovakia). The EO was sourced from Egypt and was stored in darkness at 4 °C for the duration of the analysis.

### 2.2. GC and GC/MS Examination

PGEO volatile chemical composition was determined using an Agilent Technologies 6890N gas chromatograph interfaced with a quadrupole mass spectrometer (Agilent Technologies, Santa Clara, CA, USA). An HP-5MS capillary column (30 m × 0.25 mm × 0.25 µm) was used to separate volatile components. The Agilent Technologies gas chromatograph was run using HP Enhanced ChemStation software D.03.00.611. A 10% EO solution in hexane was injected at a volume of 1 µL, and helium 5.0 was utilized as the carrier gas at a rate of 1 mL/min. The MS quadruple, split/splitless injector, and MS source were maintained at temperatures of 230 °C, 280 °C, and 150 °C, respectively. The mass scan range was 35–550 amu at 70 eV, and the split ratio was 40.8:1. The total run time was 57 min, and temperature was programed as follows: 50 °C to 75 °C (increasing rate, 3 °C/min) held for 4 min, 75 °C to 120 °C (increasing rate, 5 °C/min) held for 2 min, and 120 °C to 290 °C (increasing rate, 5 °C/min)**.** For the examination of the EO sample, the solvent delay time was 3.20 min; however, for the n-alkanes (C_7_–C_35_), it was 2.10 min.

The volatiles were identified by comparing the retention indices (RIs) of compounds found in the PGEO sample with the RIs of the n-alkane (C_7_–C_35_) series [24]. Additionally, the identification of the volatiles was carried out by comparing their spectral data with reference spectra available in the literature and in the MS library (Wiley7Nist), integrated in the HP Enhanced ChemStation program. Using the same HP-5MS capillary column, GC-FID was used to semi-quantify each component, taking into account levels greater than 0.1%.

### 2.3. Antioxidant Activity

To assess the antioxidant activity of PGEO, two standard assays were employed. In the DPPH assay, a DPPH^•^ stock solution (Sigma-Aldrich, Schnelldorf, Germany) was prepared in methanol. The stock solution was then diluted to achieve an absorbance of 0.8 at 515 nm using methanol (Uvasol^®^ for spectroscopy, Merck, Darmstadt, Germany) [25]. For the ABTS assay, the ABTS^•+^ was generated following a previously described procedure [26]. Prior to analysis, the prepared radical cation was diluted to an absorbance value of 0.7 at 744 nm using methanol. In a 96-well microtiter plate, 190 μL of the prepared solutions (DPPH radical or ABTS radical cation) was added. Subsequently, PGEO was introduced to the plates at a volume of 10 μL, achieving final concentrations ranging from 3.0 mg/mL to 0.1875 mg/mL (in methanol). Methanol served as the blank solution, and Trolox was utilized as the reference compound. The reference compound was dissolved in methanol to yield final concentrations in the wells ranging from 3.0 to 0.015 mg/mL. The reaction mixtures were shaken at 1000 rpm for 30 min at room temperature in the dark. Subsequently, absorbance was measured using a microplate reader (Glomax, Promega Inc., Madison, WI, USA) at 515 nm for the DPPH test and 744 nm for the ABTS assay. The determination of antioxidant activity was conducted relative to the standard reference Trolox (Sigma-Aldrich, Schnelldorf, Germany) that was dissolved in methanol (Uvasol^®^ for spectroscopy, Merck, Darmstadt, Germany). The overall antioxidant activity was then quantified based on a calibration curve. Each measurement was conducted three times. The TEAC and IC_50_ values for the total radical scavenging capability in both assays were reported as mean values ± standard deviation (SD).

### 2.4. Test for Antimicrobials

#### 2.4.1. Microorganisms

In our experiments to assess the antibacterial efficacy of the EO, the following bacteria were utilized: Gram-positive (G^+^) strains, including *Priestia* (formerly *Bacillus*) *megaterium* CCM 2007, *Streptococcus constellatus* CCM 4043, and *Enterococcus faecalis* CCM 4224, and Gram-negative (G^−^) strains, including *Citrobacter freundii* CCM 7187, *Shigella sonnei* CCM 4421, *Escherichia coli* CCM 3954, and *Serratia marcescens* CCM 8587. All G^+^ and G^−^ bacterial species were obtained from the Czech Collection of Microorganisms in Brno, Czech Republic. For assessing antibiofilm activity, biofilm-forming G^−^ *Salmonella enterica* was extracted and sequenced from milk production. The bacterial inoculums were cultured for 24 h at 37 °C in Mueller–Hinton Broth (MHB, Oxoid, Basingstoke, UK) before analysis. The optical density of the bacterial inoculums used was adjusted to 0.5 of the McFarland standard on the day of the experiment.

#### 2.4.2. Disk Diffusion Method

A disk diffusion susceptibility test was conducted using the aforementioned microbial strains. Mueller–Hinton Agar (MHA; Merck, Darmstadt, Germany) was inoculated with the prepared bacterial strains in 0.1 mL of Mueller–Hinton Broth (MHB). Blank disks with a diameter of 6 mm were impregnated with 10 µL of the tested EO, and then, positioned on the agar surface. Bacterial cultures were incubated at 37 °C. After a 24 h incubation period, the inhibitory activity was measured, and the results were recorded in millimeters. For G^−^ and G^+^ bacteria, positive controls included the well-known antibiotics (ATB) cefoxitin (30 µg/disk, Oxoid, Basingstoke, UK) and gentamicin. The experiment was run three times for validation.

#### 2.4.3. MIC Assay

The minimal inhibitory concentration values (MIC50 and MIC90) were determined using a previously published methodology [27]. In summary, a 96-well microtiter plate was filled with 50 μL of microbial inoculum. Subsequently, the EO was added in varying concentrations (ranging from 10 mg/mL to 0.00488 mg/mL in Mueller–Hinton Broth). Negative control wells containing Mueller–Hinton Broth with EO at the corresponding concentration were created, and positive control wells containing Mueller–Hinton Broth with the inoculum were included for maximal growth. The prepared plates were then incubated at 37 °C for a full day. Finally, absorbance at 570 nm was detected using a spectrophotometer (Glomax, Promega Inc., Madison, WI, USA). The results are expressed as MIC50 values, representing the lowest PGEO concentration required to inhibit 50% of bacterial growth, and MIC90 values, representing the lowest PGEO concentration required to inhibit 90% of bacterial growth. The test was replicated three times, and the results are presented as mean values ± standard deviation (SD).

### 2.5. In Situ Analysis on a Food Model

The antibacterial properties of PGEO were assessed *in situ* using strains of both G^+^ and G^−^ bacteria. To simulate real-world conditions, commercially available food items, such as apples and carrots, were employed as substrates for bacterial growth. The methodology applied in this study has been previously reported [28]. Apples and carrots were cut into 0.5 mm pieces with a sterile knife, dried, and cleaned with distilled water. Subsequently, 60 mm Petri plates containing the prepared substrates on agar were inoculated with bacteria. The PGEO sample under investigation was applied to sterile filter paper after being dissolved in ethyl acetate at concentrations of 500, 250, 125, and 62.5 mg/L. As a control, filter sheets exposed only to ethyl acetate were used. The filter paper with the treatment or control was placed on the lid of the Petri dish and left for a minute to allow any remaining ethyl acetate to evaporate before closing the dish. The next step involved incubating the prepared Petri dishes for seven days at 37 °C. Conventional methods were employed to measure the growth of bacteria *in situ*. The volumetric density of bacterial colonies (vv) was determined using the ImageJ program 1.8.0 from the National Institutes of Health in Bethesda, MD, USA. The volumetric density of bacterial colonies was estimated as follows:vv (%) = P/p
where P represents the stereological grid points that hit the colonies, whereas p represents the points of the stereological grid falling to the reference space (growth substrate used).

The effects of the EO vapor phase are presented as the percentage (%) of bacterial growth inhibition (BGI):BGI = [(C − T)/C] × 100(1)
where C corresponds to the control group, while T indicates the treatment group. Both groups represent bacterial growth expressed as *v*/*v*. Results obtained as negative values correspond to growth stimulation.

### 2.6. Biofilm Development Assay

Using a Bruker Daltonics MALDI-TOF MicroFlex (Bruker Daltonics, Bremen, Germany), the breakdown of proteins during the biofilm-formation process was examined. In this experiment, 20 mL of Mueller–Hinton Broth (MHB) and 100 μL of *Salmonella enterica* biofilm-forming bacterial inoculum were introduced into 50 mL polypropylene tubes. Microscopic stainless steel and plastic slides were then added to the tubes. PGEO was included in the experimental tubes at a final concentration of 0.1%, while untreated tubes served as controls. The prepared polypropylene tubes were incubated at 37 °C for 3, 5, 7, 9, 12, and 14 days with agitation at 170 rpm. Biofilms formed on the tested surfaces (steel and plastic) were removed daily using a sterile cotton swab and immediately deposited on the target plate. Additionally, planktonic cells from the control samples without EO were also examined. In the control samples, 300 µL of bacterial suspension in the culture media was added to a 2 mL Eppendorf tube (Eppendorf, Prague, Czech republic), which was then centrifuged for one minute at 12,000 rpm. The resulting pellets were washed in ultrapure water and centrifuged three times. For testing, 1 μL of planktonic cells (pellets) was placed on the target plate after reconstitution in ultrapure water. Furthermore, swabs collected from the tested surfaces and planktonic cells on the plate were covered with 1 μL of a 10 mg/mL α-cyano-4-hydroxycinnamic acid matrix. After placing a dried plate in the MALDI-TOF spectrometer, spectra were obtained. Protein spectral data were acquired in linear positive mode, with the mass-to-charge ratio adjusted between 200 and 2000. Eighteen standard global spectra (MSP) were generated through automated analysis using the Euclidean Distance Formula, and dendrograms were created using the generated MSPs [29].

### 2.7. PGEO’s Insecticidal Properties

To evaluate the impact of PGEO on *Harmonia axyridis* insecticidal activity, insect subjects were distributed among several Petri plates. A sterile filter paper was positioned in the lid of each Petri dish. Various concentrations (50, 25, 12, 5, 6.25, and 3.125%) of PGEO, diluted with a 0.1% polysorbate solution, and a 0.1% polysorbate solution alone (control) in a 100 µL volume, were applied to the filter paper. The Petri dishes were sealed with parafilm, kept at room temperature, and monitored for 24 h. After this 24 h period, the impact of PGEO on the viability of *Harmonia axyridis* was assessed.

### 2.8. Statistical Data Evaluation

Data processing was carried out using SAS^®^ software version 8. Logit analysis was performed to determine the MIC value, or the concentration at which bacterial growth was 50% and 90% inhibited.

## 3. Results

### 3.1. GC and GC/MS Analyses of P. graveolens EO Volatile Composition

Identifying the bioactive components in mixes, such as EOs, can reveal important details about their possible uses. Given that a variety of circumstances can trigger plants to produce volatile compounds, examining these constituents is an essential first step. The results obtained by using GC/MS analysis are presented in Table 1. Together with the literature and experimentally established RIs, the analysis data are expressed as a percentage of the identified components and grouped by means of the different classes of compounds. A total of 54 components were identified, representing 99.2% of the oil composition. The examined sample of PGEO was primarily distinguished by a large abundance of monoterpene compounds (79.2%), among which oxygenated monoterpenes predominated with a percentage of 78.6%. Additionally, alcohols (55.3%), and esters (18.9%) of the monoterpene class, were found in high amounts. The main constituent of EO was found to be the monoterpene alcohol β-citronellol with an abundance of 29.7%, followed by geraniol (14.6%), menthol (6.7%), and linalool (3.8%). From the class of monoterpene esters, citronellyl formate (8.5%), citronellyl pentanoate (6.0%), and geraniol formate (3.1%) were detected in notably high amounts. The remaining 48 compounds were identified in amounts of <2.5%.

### 3.2. Antioxidant Activity

The ability of PGEO to neutralize stable DPPH radicals and ABTS radicals was evaluated based on IC_50_ and TEAC values. In the DPPH assay, the IC_50_ value was found to be 1.14 ± 0.08 mg/mL, while the TEAC value was evaluated at 0.0040 ± 0.0002. For the ABTS assay, the IC_50_ value was determined to be 0.26 ± 0.02 mg/mL, while the TEAC value was estimated to be 0.0064 ± 0.0003. The obtained results clearly indicate the superior ability of PGEO to neutralize the ABTS^•+^ compared to DPPH^•^.

### 3.3. Evaluation of In Vitro Antibacterial Activity

This study applied the two most used approaches to assess the antibacterial properties of PGEO. Consequently, the disk diffusion susceptibility experiment was used to screen PGEO, and the findings are shown in Table 2. According to the data, G^+^ bacteria were more susceptible to PGEO than G^−^ bacteria. Out of the G^+^ species, *P. megaterium* showed the highest susceptibility to PGEO treatment with an inhibition zone of 14.67 ± 0.58 mm, while *S. constellatus* was slightly less sensitive (8.33 ± 0.58 mm), and *E. faecalis* was the most resistant (2.33 ± 0.58 mm). Taking into account G^−^ strains, the obtained data show that the observed inhibition zones obtained in the PGEO treatment ranged from 2.33 ± 0.58 mm for the most resistant *S. marcescens* and *C. freundii* to 11.67 ± 0.58 mm for the most sensitive *S. sonnei*. However, considering the G^−^ strain, PGEO showed the highest activity rates towards the biofilm-forming bacterium *S. enterica*, with a displayed inhibition zone of 14.33 ± 0.58 mm.

The positive antibacterial properties of the PGEO mixture, as indicated by the results from the disk diffusion method, prompted further antimicrobial research. The MIC test was conducted, and Table 3 summarizes the obtained findings. Upon analysis, it was confirmed that overall, PGEO exhibited better activity towards G^+^ bacterial strains. The treatment of G^+^ strains with PGEO showed the lowest MIC 50 (0.333 ± 0.091 mg/mL) and MIC90 (0.387 ± 0.083 mg/mL) values in inhibiting *E. faecalis*, and the highest in inhibiting *P. megaterium* (MIC 50 of 0.680 ± 0.202 mg/mL and MIC90 of 0.790 ± 0.210 mg/mL). Among the G^−^ bacteria, *E. coli* was the most susceptible to the effects of PGEO, with MIC 50 and MIC 90 values of 1.443 ± 0.110 mg/mL and 1.570 ± 0.105 mg/mL, respectively. Conversely, *C. freundii* demonstrated the highest resistance to PGEO, with an MIC 50 value of 3.310 ± 0.070 mg/mL and an MIC90 value of 3.523 ± 0.047 mg/mL, along with *S. sonnei*, showing an MIC50 value of 3.297 ± 0.055 mg/mL and an MIC90 value of 3.450 ± 0.101 mg/mL. However, in the case of the G^−^ biofilm-forming strain, *S. enterica*, it was found to be more sensitive to the effects of PGEO. Treatment with PGEO resulted in 50% inhibition of this bacterium at a concentration of 0.157 ± 0.006 mg/mL, while 90% inhibition was achieved with a PGEO concentration of 0.169 ± 0.080 mg/mL.

From the presented results, it may be concluded that PGEO has a strong inhibitory effect on the studied G^+^ bacterial species. However, its effectiveness towards G^−^ strains is limited, except for the more susceptible, biofilm-forming *S. enterica*. Additionally, the antibiotic resistance evaluation conducted by using the disk diffusion method shows stronger activity of the tested antibiotics compared to the effects of PGEO.

### 3.4. In Situ Antibacterial Activity Assessment

Since the tested EO showed promising antibacterial effects, in the next step, this study was designed to analyze its antibacterial impact in the vapor phase. PGEO’s effects were assessed against G^+^, G^−^, and biofilm-forming G^−^ bacteria growing on apples and carrots (Table 4).

In an apple model infected with G^+^ bacteria, PGEO was the most effective against *E. faecalis* (95.67 ± 4.32%) at a concentration of 62.5 µg/mL, whereas in the inhibition of *P. megaterium*, it showed pro-bacterial effects at all applied concentrations. The G^−^ bacterial strain vapor phase of PGEO was the most effective at a concentration of 500 µg/mL in inhibiting the growth of *S. marcescens* (74.23 ± 3.72%), and at a concentration of 62.5 µg/mL in inhibiting the *E. coli* strain (67.54 ± 3.56%). However, pro-bacterial activity of PGEO was noted against *C. freundii* and *S. sonnei* at an applied concentration of 500 µg/mL (−23.72 ± 3.73% resp. −34.56 ± 3.24%).

Considering the obtained results for the inhibition of bacterial growth of the G^+^ strains on the carrot model, PGEO was the most effective in the inhibition of *P. megaterium* (96.58 ± 3.57%) at a concentration of 62.5 µg/mL, whereas *S. consellatus* (89.56 ± 4.35%) and *E. faecalis* (65.74 ± 3.63%) were maximally inhibited at the applied concentration of 500 µg/mL. For G^−^ strains, the PGEO vapor phase was the most effective at the lowest applied concentration (62.5 µg/mL), with observed inhibition effects of 78.95 ± 3.42%, 97.86 ± 3.42%, and 87.56 ± 3.63% towards *S. marcescens, S. sonnei,* and *E. coli*, respectively, growing on the carrot model.

Summarizing the results obtained for the inhibition of biofilm-forming *S. enterica*, PGEO effectively reduced its growth on the carrot model at a concentration of 250 µg/mL to the extent of 96.23 ± 3.72%. Against this bacterium growing on an apple model, PGEO showed modest effects with maximum efficiency at an applied concentration of 500 µg/mL.

### 3.5. Antibiofilm Activity of PGEO against Salmonella enterica

This research was extended to investigate the inhibitory potential of PGEO in the development of *Salmonella enterica* biofilm on different surfaces, given its demonstrated effectiveness. Figure 1 depicts the results of assessing the effects of PGEO against biofilm-producing *S. enterica* growing on plastic and stainless steel, utilizing a MALDI-TOF MS Biotyper. Molecular changes in the biofilm were systematically compared with planktonic cells, serving as a control, as the spectra of the control groups exhibited consistent alterations. It is noteworthy that the control groups included data from planktonic cells and biofilm spectra that had not been exposed to the effects of the tested essential oil (spectra not presented).

The acquired data unequivocally demonstrate that the influence of PGEO on biofilm production within the experimental groups was evident right from the initiation of the trial. Notably, the spectra of the two experimental surfaces (plastic and stainless steel) distinctly deviated from the control planktonic spectra, indicating the discernible impact of the treatment. Commencing from the third day of the experiment, substantial alterations in the protein profile became apparent, signifying a disruption in biofilm formation within the experimental groups, as evidenced by the evolving spectral records. Specifically, on the third day (3SEP, 3SES, 3PC), the mass spectra of the experimental group continued to evolve distinctively from the control planktonic spectrum. These observed changes in the protein profile within the experimental groups are indicative of biofilm disintegration. This consistent pattern persisted on the seventh day of the trial. On the ninth day of the experiment, the mass spectrum of the plastic surface exhibited sustained disparity, while the stainless-steel experimental group displayed a return to similarity with the control group. Nevertheless, a subsequent return to dissimilarity in the mass spectra of both the experimental and control planktonic groups on both surfaces was noted during the final two days of the experiment. Comprehensive interpretation of the data underscores that PGEO exerts a pronounced effect on disrupting the homeostasis of *S. enterica* biofilm, leading to the suppression of microorganism growth on both experimental surfaces from the outset of the anticipated biofilm formation.

Furthermore, a dendrogram was constructed using the presented data (Figure 2). The MSP distances between the planktonic cells and controls exhibited the shortest spans, with an observable increase in MSP distances for the experimental group over the course of the trial. Notably, the experimental groups displayed the shortest MSP distances on the third day of the experiment. Conversely, on the fourteenth day, the experimental group showcased the longest MSP distance, prominently influencing the plastic surface. However, on the ninth day, the MSP distance measured from the stainless-steel surface of the experimental group decreased. These results collectively indicate that PGEO manifests detrimental and inhibitory effects on *S. enterica* biofilm formation on both plastic and stainless-steel surfaces.

### 3.6. Insecticidal Activity

The results obtained for the assessed insecticidal activity of PGEO against *Harmonia axyridis* are displayed in Table 5. The presented results indicate the maximum insecticidal action of the tested EO at applied concentrations of 100% and 50%. However, when applied to *H. axyridis*, at concentrations of 6.25% and 3.125%, PGEO did not show strong repellent properties. In *H. axyridis*, 50% of the population was impacted by PGEO at a 25% concentration, and 12.5% of PGEO was active for 33.33% of the insects. 

## 4. Discussion

Numerous EOs have demonstrated substantial *in vitro* and *in situ* efficacy against spoilage organisms and foodborne pathogens. However, prior to their implementation in commercial settings, it is imperative to investigate potential synergies between EOs and other chemicals, as well as their compatibility with various processing practices. The primary objective of this study was to assess the insecticidal effects of the essential oil extracted from *Pelargonium graveolens*, considering its chemical composition, antioxidative activity, and antibacterial potential against G^+^, G^−^, and biofilm-forming bacteria, both *in vitro* and *in situ*.

In this investigation, by employing the GC/MS technique, 54 components were successfully identified as constituents of the essential oil. The monoterpene alcohols *β*-citronellol and geraniol are the components present at the highest concentrations. According to our data, in the study of Bigos et al. [11] the primary constituents of *Pelargonium graveolens* Ait. were found to be citronellol (26.7%) and geraniol (13.4%). Other common compounds found in geranium EO included nerol (8.7%), citronellyl formate (7.1%), isomenthone (6.3%), linalool (5.2%), and 10-*epi-γ*-eudesmol (4.4%), among the sixty-seven constituents discovered. Moreover, the chemical composition of PGEO was studied in many other publications [24,30], however significant variations in the chemical composition can be noted, particularly depending on the origin of the plant used for EO extraction. These variations in composition can be attributed to diverse climatic and environmental factors. Notably, *Pelargonium graveolens* essential oil (PGEO) obtained from Iran exhibited citronellol (48.44%), octen-1-ol (18.61%), and geraniol (9.70%) as predominant components [31]. Conversely, PGEO from Tunisia demonstrated significant concentrations of *β*-citronellol (21.90%), citronellyl formate (13.20%), and geraniol (11.10%). In the case of PGEO from Serbia, the main constituents comprised citronellol (24.54%), geraniol (15.33%), citronellyl formate (10.66%), and linalool (9.80%) [10].

Numerous protocols have been developed to evaluate the free radical scavenging activity of the EOs due to their chemical composition complexity [32,33]. Nevertheless, the most commonly utilized assays are the DPPH and ABTS methods. In our investigations, we assessed the antioxidant capacity of PGEO. Our findings align with previous reports, emphasizing the superiority of ABTS neutralization for examining plants containing both hydrophilic and lipophilic compounds [34]. Furthermore, it was demonstrated by Boukhris et al. [35] that the antioxidant activity of the EOs could be partially attributed to the presence of the main compounds (*β*-citronellol and geraniol). These findings showed that during the full blooming phase, PGEOs have higher antioxidant capacity than PGEOs during the early flowering and dormant stages. This intriguing antioxidant activity is consistent with data gathered from other surveys [36,37].

The literature data consistently indicate that G^+^ species tend to be more susceptible to EO exposure compared to G^−^ ones. However, further investigations are required to gather comprehensive data on the efficacy of various essential oils in distinct food matrices. Consistent with the existing literature, our study underscores that G^+^ strains were the most sensitive bacteria when exposed to PGEO. An exception was the strong inhibition power of PGEO towards the biofilm-forming bacterium *S. enterica*, which is a representative of the G^−^ species. Previous reports by Al-Mijalli et al. [38] showed a strong antibacterial impact of PGEO against all tested strains except *Salmonella typhimurium*. In this research, G^+^ strains were also more sensitive to the effects of PGEO. Moreover, in this research, the authors concluded that the antibacterial activity of the investigated volatile oils was, on average, lower when compared to the reference antimicrobial compounds, which agrees with the results obtained in this study. In another study, Hsouna and Hamidi [4] showed strong inhibition power of PGEO against bacterial species, especially towards G^+^ species, where *B. cereus* ATCC 14579 and *S. aureus* ATCC 25923 were the most sensitive, with inhibition zone diameters of 26 ± 0.02 mm and 24 ± 0.3 mm, respectively. However, a difference in data obtained from earlier studies on the antibacterial action of PGEOs can be noted [39,40]. The greatest antibacterial potency of this EO was discovered towards G^−^ *P. aeruginosa* by El Asbahani et al. [40]. In the work of Ghannadi et al. [39], EO obtained from *P. graveolens* showed greater effectiveness towards *S. aureus* and *P. aeruginosa* compared to the tested reference antibiotics. Studies conducted by Ghannadi et al. [39] and Hsouna and Hamdi [4] reported that PGEO from Iran and Tunisia, respectively, did not exhibit any impact against a foodborne reference strain of *Listeria monocytogenes* PTCC 1297 and *Listeria monocytogenes* 12228.

Incorporating fresh fruits and vegetables into one’s diet is recognized as an integral component of a healthy lifestyle. Conversely, diminished consumption of these food items has been associated with adverse health outcomes and an increased risk of developing certain non-communicable diseases. Throughout various stages of the production process, fresh vegetables are occasionally susceptible to contamination with foodborne pathogenic bacteria. Pathogenic bacteria can enter a field through polluted water, soil, livestock, wildlife, and equipment used during harvest, as well as through cross-contamination from agricultural workers [41]. One of the solutions in the fight against food contamination can be found in the use EOs, considering that these mixtures are mostly used as flavorings in the food sector. In our study, we demonstrated the effects of PFEO against eight bacteria inoculated on apple and carrot food models. Our results show that PGEO was more effective, especially at the lowest concentration. Todd et al. [42] inoculated *Salmonella* into organic romaine, iceberg lettuce, and organic baby and mature spinach. In this study, the direct contact method was used to evaluate the effects of cinnamon leaf EO, and the greatest decreases in *Salmonella* growth were obtained with higher doses and longer treatment times. In another study, *Salmonella* Enteritidis, *E. coli*, and *L. monocytogenes* grown in fresh leafy vegetables were treated with oregano and rosemary EOs by Medeiros Barbosa et al. [43]. Following a 5 min treatment, oregano EO exhibited the most pronounced inhibitory effect, with a minimum inhibitory concentration (MIC) of 0.6 μL/mL and a reduction of ≥3 log cycles observed in all bacteria. Furthermore, in a study conducted by de Azeredo et al. [44] iceberg lettuce, beets, and rocket inoculated with *Listeria monocytogenes*, *Yersinia enterocolitica*, *Aeromonas hydrophila*, and *Pseudomonas fluorescens* were subjected to treatment with oregano and rosemary EOs. The bacterial counts were most significantly reduced by oregano EO at an MIC of 1.25 to 5 μL/mL, from an initial value of around 8 log CFU/g to <1.0 to 2.7 log CFU/g. Additionally, oregano EO by itself produced the greatest decrease in natural microbiota counts. The findings of Kačániová et al. [28] demonstrated the potent antibacterial activity of cedar EO in inhibiting the development of *M. luteus* and *S. marcescens* on models of bread, carrots, and celery using the same method as that used in this study. According to these findings, cedar EO exhibited antifungal action against *Penicillium expansum, P. chrysogenum, P. italicum*, and *P. aurantiogriseum* which were developing on the bread model. When evaluated on carrots and celery as substrates, the highest doses of the EO exhibited potent antifungal activity, effectively inhibiting the growth of all tested fungal strains.

As mentioned, the formation of microbial biofilms can interrupt industrial processes and consequently represents a significant problem for the food processing sector. A complex matrix of microorganisms with cells bound to a biotic or abiotic surface is called a biofilm. Currently, there are very few substances that have been shown to exhibit activity on biofilms. As a result, novel antibiofilm compounds are required. Fortunately, certain EOs have been shown to be successful in the fight against antifungal and antibacterial biofilms. The activity of tea tree essential oil (TTEO), lavender essential oil (LEO), melissa essential oil (MEO), and lemon essential oil (LEO) on biofilms generated by reference strains of *S. aureus* and *E. coli* is presented in a study conducted by Budzyűska et al. [20]. The results obtained in this study indicated that compared to LEO and its primary constituents, linalyl acetate and linalool, MEO demonstrated a greater antibiofilm impact. In contrast to the prevailing notion that G^−^ bacteria exhibit greater resistance to EOs, the conducted tests revealed that the biofilm formed by *E. coli* was more susceptible than *S. aureus* biofilms to the influence of EOs, particularly TTEO, which effectively eradicated it following 1 h of exposure to a concentration of 0.78%. Unlike LEO and TTEO, the impact of MEO demonstrated a more time-dependent nature. The findings gleaned from this study suggest that PGEO possesses the capability to disrupt the biofilm formation of *S. enterica*, a phenomenon evident from the initial stages of the experiment. The influence of the treatment was discerned through disparities in the spectra of this bacterium growing on plastic and stainless steel compared to the control planktonic spectra. A parallel observation was reported in the study by Kačániová et al. [28], utilizing a MALDI-TOF MS Biotyper to evaluate the effects of cedar essential oil against *Pseudomonas fluorescens* and *Salmonella enterica*, both biofilm-producing bacteria. In both cases, the bacterial biofilms treated with cedar essential oil exhibited alterations in their protein profiles compared to the control spectra.

Considering that EOs may be used in both conventional and organic agriculture with little or no environmental impact, their use as bioinsecticides is of great interest [45]. Although EOs have a complex chemical composition and can act on multiple sites, most of the paralysis and death of insects caused by these mixtures is explained by damage to the central nervous system [46]. From the literature data, it has been observed that *P. graveolens* EO, commonly referred to as geranium EO, has insecticidal and insect-repelling qualities. According to previous reports, *P. graveolens* and its constituents have insecticidal activity against a variety of pests, including the sweet potato whitefly (*Bemisia tabaci* Gennadius) [47], the house fly (*Musca domestica* L.) [48], the Japanese termite (*Reticulitermes speratus* Kolbe) [49], and the maize weevil (*Sitophilus zeamais* Motschulsky) [50]. In our investigation, geranium EO had a strong repellent effect, especially at doses of 100% and 50%, suggesting that it has considerable promise as a *Harmonia axyridis* repellent. As previously reported, geranium EO has a repellent effect on both males and females. This work supports previous reports, as the EO demonstrated deterrent and repellent effects in multiple-choice and repellent tests [51,52].

## 5. Conclusions

This investigation systematically explored the chemical composition, antioxidant attributes, antimicrobial properties (both *in vitro* and *in situ*), antibiofilm efficacy, and insecticidal characteristics of a commercially obtained *Pelargonium graveolens* essential oil (PGEO) sourced from the Hanus Company in Slovakia. G^−^ and G^+^ bacteria were both susceptible to the antibacterial action of PGEO. In contrast to the harmful health effects linked to the intake of synthetic antimicrobials, the antioxidant qualities of this essential oil may also benefit consumers. Therefore, PGEO may be a good option for research and development as a substitute natural antibacterial to stop both Gram-positive and Gram-negative bacteria from contaminating food items. This study’s findings showed that PGEO has antibacterial effectiveness against pathogens that are foodborne and are cultured in lab environments. To compare the efficacy of PGEO in blocking foodborne microorganisms, more research is necessary to examine the potential effects of PGEO, either by itself or in conjunction with different physical treatments. In conclusion, plant-derived EOs can be considered a valid alternative to chemical pesticides as some of them have proven effective in controlling harmful insects.

## Figures and Tables

**Figure 1 foods-13-00033-f001:**
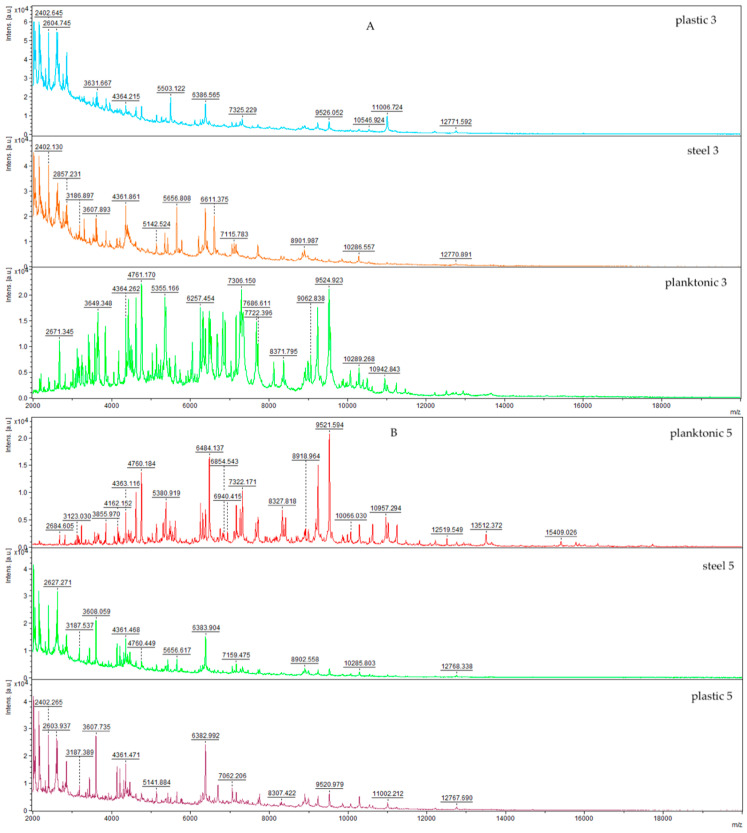
MALDI-TOF mass spectra of *S. enterica* during the development of the biofilm: (**A**) 3rd day; (**B**) 5th day; (**C**) 7th day; (**D**) 9th day; (**E**) 12th day; (**F**) 14th day.

**Figure 2 foods-13-00033-f002:**
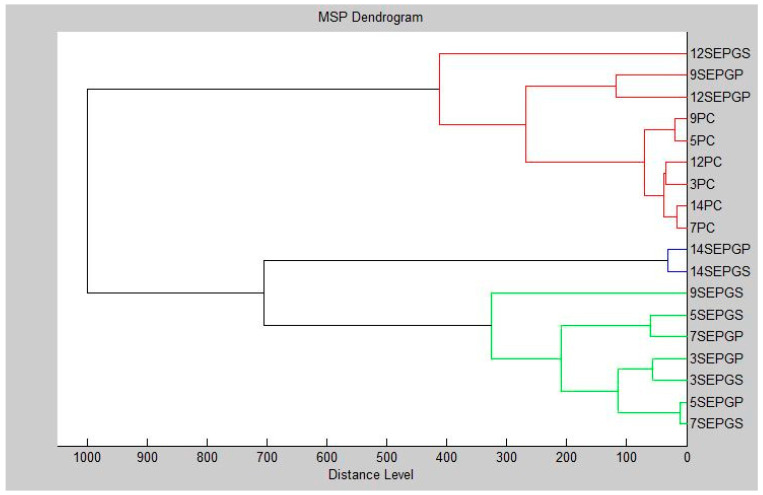
Dendrogram of *S. enterica* biofilm progress after PGEO exposition: SE—*Salmonella enterica*; PC—planktonic cell; P—plastic; S—stainless steel.

**Table 1 foods-13-00033-t001:** Chemical composition of PGEO.

No	RI (lit.)	RI (calc.) ^a^	Compound ^b^	%
			monoterpenes	80.9
			monoterpene hydrocarbons	0.6
1	939	934	α-pinene	0.5
2	990	988	β-myrcene	tr ^c^
3	1002	1006	α-phellandrene	tr
4	1024	1027	p-cymene	tr
5	1029	1032	limonene	0.1
			oxygenated monoterpenes	80.3
			monoterpene epoxides	1.8
6	1073	1075	trans-dihydro-rose oxide	0.2
7	1108	1111	cis-rose oxide	1.2
8	1125	1130	trans-rose oxide	0.4
			monoterpene alcohols	55.3
9	1096	1100	linalool	3.8
10	1171	1170	menthol	6.7
11	1182	1189	iso-menthol	0.2
12	1188	1193	α-terpineol	0.3
13	1225	1232	β-citronellol	29.7
14	1252	1253	geraniol	14.6
			monoterpene aldehydes	1.5
15	1153	1157	citronellal	0.1
16	1238	1239	neral	0.6
17	1267	1267	geranial	0.8
			monoterpene ketones	1.1
18	1162	1160	iso-menthone	1.1
			monoterpene esters	20.6
19	1273	1273	citronellyl formate	8.5
20	1298	1297	geraniol formate	3.1
21	1352	1352	citronellyl acetate	1.1
22	1446	1445	citronellyl propionate	0.2
23	1625	1625	citronellyl pentanoate	6.0
24	1656	1657	geranyl valerate	0.7
25	1696	1695	geranyl tiglate	1.0
			sesquiterpenes	17.2
			sesquiterpene hydrocarbons	15.7
26	1351	1349	α-cubebene	0.2
27	1376	1376	α-copaene	0.7
28	1388	1384	β-bourbonene	1.8
29	1419	1419	trans-caryophyllene	1.6
30	1434	1430	trans- α-bergamotene	0.1
31	1441	1441	aromadendrene	0.8
32	1451	1450	amorpha-4,11-diene	0.6
33	1454	1456	α-humulene	0.3
34	1460	1460	allo-aromadendrene	0.3
35	1466	1470	cis-muurola-4(14),5-diene	0.7
36	1479	1472	γ-muurolene	0.3
37	1481	1481	germacrene D	2.2
38	1484	1484	α-amorphene	0.2
39	1490	1490	β-selinene	0.6
40	1493	1494	trans-muurola-4(14),5-diene	0.8
41	1500	1496	α-muurolene	0.2
42	1505	1502	(E,E)-α-farnesene	0.2
43	1505	1506	β-bisabolene	0.4
44	1513	1512	γ-cadinene	1.8
45	1522	1521	trans-calamenene	0.6
46	1523	1525	δ-cadinene	0.8
47	1534	1533	trans-cadina-1,4-diene	0.5
			oxygenated sesquiterpenes	1.5
			sesquiterpene ethers	0.3
48	1550	1549	α-agarofuran	0.3
			sesquiterpene alcohols	1.2
49	1578	1577	spathulenol	0.2
50	1628	1629	1-epi-cubenol	0.2
51	1646	1644	cubenol	0.2
52	1646	1646	α-muurolol	0.3
53	1654	1652	α-cadinol	0.3
			non-terpenic compounds	1.1
			esters	1.1
54	1585	1583	2-phenyl ethyl tiglate	1.1
			total	99.2

^a^ Values of retention indices on HP-5MS column; ^b^ identified compounds; ^c^ tr compounds identified in amounts of less than 0.1%.

**Table 2 foods-13-00033-t002:** Antimicrobial activity of PGEO obtained by disk diffusion assay displayed in mm.

Microorganism	Inhibition Zone	ATB
Gram-positive bacteria		
*Enterococcus faecalis*	2.33 ± 0.58	11.33 ± 0.58
*Streptococcus constellatus*	8.33 ± 0.58	24.33 ± 0.58
*Priestia* *megaterium*	14.67 ± 0.58	22.33 ± 0.58
Gram-negative bacteria		
*Serratia marcescens*	2.33 ± 0.58	19.67 ± 0.58
*Citrobacter freundii*	2.33 ± 0.58	24.67 ± 0.58
*Shigella sonnei*	11.67 ± 0.58	23.33 ± 0.58
*Escherichia coli*	2.67 ± 0.58	19.33 ± 0.58
Biofilm-forming bacteria		
*Salmonella enterica*	14.33 ± 0.58	22.33 ± 0.58

Antibiotics used as controls are the following: cefoxitin for G^−^ bacteria, and gentamicin for G^+^ bacteria. Inhibition zones are presented in mm.

**Table 3 foods-13-00033-t003:** Minimal inhibitory concentrations of PGEO (mg/mL).

Microorganism	MIC50	MIC90
Gram-positive bacteria		
*Enterococcus faecalis*	0.333 ± 0.091	0.387 ± 0.083
*Streptococcus constellatus*	0.353 ± 0.021	0.403 ± 0.031
*Priestia* *megaterium*	0.680 ± 0.202	0.790 ± 0.210
Gram-negative bacteria		
*Serratia marcescens*	1.540 ± 0.090	1.717 ± 0.071
*Citrobacter freundii*	3.310 ± 0.070	3.523 ± 0.047
*Shigella sonnei*	3.297 ± 0.055	3.450 ± 0.101
*Escherichia coli*	1.443 ± 0.110	1.570 ± 0.105
Biofilm-forming bacteria		
*Salmonella enterica*	0.157 ± 0.006	0.169 ± 0.080

**Table 4 foods-13-00033-t004:** *In situ* antibacterial analyses in vapor phase of PGEO on apples and carrots.

Food Model	Microorganisms	Inhibition of Bacterial Growth (%)			
		Concentration of EO in μg/mL			
		62.5	125	250	500
Apple					
G^+^	*Enterococcus faecalis*	95.67 ± 4.32	87.34 ± 3.23	76.53 ± 3.23	63.73 ± 3.29
	*Streptococcus constellatus*	17.43 ± 3.63	33.38 ± 3.36	56.78 ± 3.21	76.84 ± 3.78
	*Priestia* *megaterium*	−12.47 ± 4.67	−9.34 ± 5.78	−35.76 ± 4.67	−6.76 ± 3.27
G^−^	*Serratia marcescens*	36.75 ± 2.83	45.72 ± 4.29	57.84 ± 3.61	74.23 ± 3.72
	*Citrobacter freundii*	34.45 ± 3.43	24.62 ± 3.65	15.67 ± 3.61	−23.72 ± 3.73
	*Shigella sonnei*	34.56 ± 2.51	23.65 ± 3.72	12.64 ± 2.83	−34.56 ± 3.24
	*Escherichia coli*	67.54 ± 3.56	53.23 ± 4.23	32.56 ± 3.84	26.74 ± 3.67
BFB	*Salmonella enterica*	23.45 ± 5.23	35.78 ± 3.21	45.28 ± 3.62	58.93 ± 3.26
Carrot					
G^+^	*Enterococcus faecalis*	−9.84 ± 3.38	19.45 ± 2.86	44.67 ± 2.86	65.74 ± 3.63
	*Streptococcus constellatus*	23.56 ± 2.73	45.67 ± 2.95	67.89 ± 3.62	89.56 ± 4.35
	*Priestia* *megaterium*	96.58 ± 3.57	56.78 ± 4.78	47.56 ± 3.28	78.97 ± 4.26
G^−^	*Serratia marcescens*	78.95 ± 3.42	63.23 ± 2.83	54.73 ± 3.72	48.25 ± 3.42
	*Citrobacter freundii*	−24.56 ± 3.67	24.48 ± 3.48	6.78 ± 1.43	−35.47 ± 2.72
	*Shigella sonnei*	97.86 ± 3.42	56.67 ± 3.72	34.84 ± 4.38	67.56 ± 3.84
	*Escherichia coli*	87.56 ± 3.63	77.34 ± 4.63	64.56 ± 2.75	23.56 ± 5.32
BFB	*Salmonella enterica*	56.84 ± 2.52	78.67 ± 3.45	96.23 ± 3.72	−26.73 ± 4.27

**Table 5 foods-13-00033-t005:** Insecticidal activity of PGEO against *Harmonia axyridis*.

Concentration (%)	Number of Living Individuals	Number of Dead Individuals	Insecticidal Activity (%)
100	0	30	100.00
50	0	30	100.00
25	15	15	50.00
12.5	20	10	33.33
6.25	25	5	16.67
3.125	28	2	6.67
Control group	30	0	0.00

## Data Availability

Data are contained within the article.

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
