# Peer review of "An In-Depth Study on the Chemical Composition and Biological Effects of Pelargonium graveolens Essential Oil"

_foods, 2023, doi:10.3390/foods13010033_

Round 1

Reviewer 1 Report

Comments and Suggestions for Authors

The article "An in-depth study on the chemical composition and biological effects of Pelargonium graveolens essential oil" demonstrates a large amount of work. However, it should be strongly improved. Some suggestions are listed below:

Cited bibliography is complete, but can be updated

Once EO or EOs are defined, use always the abbreviated form

Lines 80-82: the writing should be revised

Line 114: replace minutes by min

Line 116: replace min. by min

Line 120: complete the citation

Line 143: through what calculations is the final result reached?

Line 149: replace Gram positive by Gram negative

Line 155: no yeasts are named in the study

Line 163: please specify that the inhibition halo is measured in mm (if applicable)

Line 243: there is an extra ","

Lines 418 and 442: revise the spaces between words

Lines 474-475: do not repeat "Penicillium"

Lines 524-526: revise the formatting

The discussion describes the results again. It is suggested to unify both sections.

The conclusions repeat the results. Adding new information would improve this section, for example, according to the results, in which products the oil under study could be applied.

Comments on the Quality of English Language

Quality of the English language should be strongly revised. Please replace informal terms, such as "We used the following bacteria in our experiments" using more formal forms of expression.

Language supervision by an expert or native speaker is recommended.

Author Response

The Authors are very grateful to the Reviewer for their valuable comments. We want to thank the Reviewer for the time devoted to point out constructive and important comments to improve our paper.

The article "An in-depth study on the chemical composition and biological effects of Pelargonium graveolens essential oil" demonstrates a large amount of work. However, it should be strongly improved. Some suggestions are listed below:

Point 1: Cited bibliography is complete but can be updated.

Response: One of reviewer recommended reducing the number of citations to 30, for this reason we would not like to add or subtract any more. All cited sources characterize individual points addressed in the experimental part of the article.

Point 2: Once EO or EOs are defined, use always the abbreviated form.

Response: It was changed.

Point 3: Lines 80-82: the writing should be revised.

Response: it was revised.

Point 4: Line 114: replace minutes by min.

Response: It was changed.

Point 5: Line 116: replace min. by min.

Response: It was changed.

Point 6: Line 120: complete the citation.

Response: It was added.

Point 7: Line 143: through what calculations is the final result reached?

Response: The determination of antioxidant activity was conducted relative to the standard reference Trolox (Sigma Aldrich, Schnelldorf, Germany) that was dissolved in methanol (Uvasol® for spectroscopy, Merck, Darmstadt, Germany). The overall antioxidant activity was then quantified based on a calibration curve. And it was added.

Point 8: Line 149: replace Gram positive by Gram negative.

Response: It was changed.

Point 9: Line 155: no yeasts are named in the study.

Response: It was deleted.

Point 10: Line 163: please specify that the inhibition halo is measured in mm (if applicable)

Response: It was changed.

Point 11: Line 243: there is an extra ","

Response: It was deleted.

Point 12: Lines 418 and 442: revise the spaces between words.

Response: It was changed.

Point 13: Lines 474-475: do not repeat "Penicillium".

Response: It was changed.

Point 14: Lines 524-526: revise the formatting.

Response: It was changed.

Point 15: The discussion describes the results again. It is suggested to unify both sections.

Response: Thank you for your comments, but with agreement of instruction of authors we prefer have discussion as a separate chapter. In the discussion, we deleted the mentioned data from the results.

Point 16: The conclusions repeat the results. Adding new information would improve this section, for example, according to the results, in which products the oil under study could be applied.

Response: The conclusion was modified by the comment.

Point 17: Quality of the English language should be strongly revised. Please replace informal terms, such as "We used the following bacteria in our experiments" using more formal forms of expression.

Language supervision by an expert or native speaker is recommended.

Response: English language was corrected.

Reviewer 2 Report

Comments and Suggestions for Authors

This manuscript on the composition and different properties of the Pelargonium graveolens essential oil. It is a comprehensive and well designed research. However, there some minor revision. For example, it would be very interesting to see the GC-MS chromatogram of the essential oil in the manuscript. Also, some minor English editing, there is no need to write " We used ...". The study itself is important and texts should be in passive form. Line 283: "The PGEO mixture's generally.... ", please rewrite it and it should be at the beginning of the sentence: The PGEO mixtures generally...

There is no need to use 52 references, 25-30 references in a normal research article is enough, no need to give a reference for well known texts.

Comments on the Quality of English Language

Some minor English editing, there is no need to write " We used ...". The study itself is important and texts should be in passive form. Line 283: "The PGEO mixture's generally.... ", please rewrite it and it should be at the beginning of the sentence: The PGEO mixtures generally...

Author Response

The Authors are very grateful to the Reviewer for their valuable comments. We want to thank the Reviewer for the time devoted to point out constructive and important comments to improve our paper.

Point 1: This manuscript on the composition and different properties of the Pelargonium graveolens essential oil. It is a comprehensive and well-designed research. However, there some minor revision. For example, it would be very interesting to see the GC-MS chromatogram of the essential oil in the manuscript. Also, some minor English editing, there is no need to write " We used ...". The study itself is important and texts should be in passive form. Line 283: "The PGEO mixture's generally.... ", please rewrite it and it should be at the beginning of the sentence: The PGEO mixtures generally...

Response: It was changed.

Point 2: There is no need to use 52 references, 25-30 references in a normal research article is enough, no need to give a reference for well known texts.

Response: All references were adequately characterizing all evaluated results. One of reviewer recommend more references, so based on knowledge with our results, we would keep a number of authors.

Point 3: Some minor English editing, there is no need to write " We used ...". The study itself is important and texts should be in passive form. Line 283: "The PGEO mixture's generally.... ", please rewrite it and it should be at the beginning of the sentence: The PGEO mixtures generally.

Response: English language was corrected.

Reviewer 3 Report

Comments and Suggestions for Authors

Dear Authors,

In general, the manuscript represents an original, completed, well-planned and methodologically good research. My overall impression is that you did a great job in the research of Pelargonium graveolens essential oils, their properties and possible application for agriculture and food industry. Experimental (Material and methods) part was described clearly; results, discussion and conclusions are presented properly. The length of the manuscript and the citation of previous publications are sufficient.

Despite all advantages (actuality, novelty, proper presentation of obtained data, etc.) of the research performed, the manuscript contains several important shortcomings, and they should be eliminated:

Lines 49-52: …essential oils (EOs), also known as odoriferous/volatile oils with antimicrobial and antioxidant effects, mainly due to global development and the emergence of numerous illnesses associated with modern civilization. Illogical, please correct.

In the 50 line you inserted abbreviation EOs of essential oils, but later (lines 61, 66...) you are using full words. Please revised all the manuscript text, regarding this remark.

Lines 80/81: Among these, there is growing interest in using (EOs) and EO as alternative agents. Revise it.

Line 183: Apple and carrot were cut into 0.5 mm pieces...Perhaps vegetables were chopped, smashed or ground...

Line 120:...GEO sample with the RI of the n-alkanes (C7–C35) series {Citation}. . Please, insert reference and revise all citations.

Table 1: a-muurolol.

Are you sure that geranyl valerate and geranyl tiglate are not sesquiterpenoids?

Line 258: compounds. identified. No dot.

Line 513: As previously reported in choice and no-choice bioassays...Unclear, what it is, comment it?

Line 518: anti-insecticidal ?

Line 247: A total of 54 components and 99.2 % of the total were identified. Too much of totals, revise it.

In Discussion section: insert links to your obtained results (Tables and Figures).

Comments on the Quality of English Language

The text is well written, it is easy to follow.  Only a minor editing of English language required.

My recommendation: moderate revision.

Author Response

In general, the manuscript represents an original, completed, well-planned and methodologically good research. My overall impression is that you did a great job in the research of Pelargonium graveolens essential oils, their properties and possible application for agriculture and food industry. Experimental (Material and methods) part was described clearly; results, discussion and conclusions are presented properly. The length of the manuscript and the citation of previous publications are sufficient.

The Authors are very grateful to the Reviewer for their valuable comments. We want to thank the Reviewer for the time devoted to point out constructive and important comments to improve our paper.

Despite all advantages (actuality, novelty, proper presentation of obtained data, etc.) of the research performed, the manuscript contains several important shortcomings, and they should be eliminated:

Point 1: Lines 49-52: …essential oils (EOs), also known as odoriferous/volatile oils with antimicrobial and antioxidant effects, mainly due to global development and the emergence of numerous illnesses associated with modern civilization. Illogical, please correct.

Response: It was revised.

Point 2: In the 50 line you inserted abbreviation EOs of essential oils, but later (lines 61, 66...) you are using full words. Please revised all the manuscript text, regarding this remark.

Response: It was changed.

Point 3: Lines 80/81: Among these, there is growing interest in using (EOs) and EO as alternative agents. Revise it.

Response: It was revised.

Point 4: Line 183: Apple and carrot were cut into 0.5 mm pieces...Perhaps vegetables were chopped, smashed or ground...

Response: Apple and carrot were cut with sterile knife.

Point 5: Line 120:...GEO sample with the RI of the n-alkanes (C7–C35) series {Citation}. . Please, insert reference and revise all citations.

Response: According to the comment, we have added the citation.

Point 6: Table 1: a-muurolol.

Response: Name of the compound is correct in the Table 1.

Point 7: Are you sure that geranyl valerate and geranyl tiglate are not sesquiterpenoids?

Response: Geraniol represents the monoterpene alcohol, so its tiglat and valerate derivatives are monoterpene esters. We apologies for the mistake.

Point 8: Line 258: compounds. identified. No dot.

Response: According to the comment, we have removed the dot.

Point 9: Line 513: As previously reported in choice and no-choice bioassays...Unclear, what it is, comment it?

Response: It was corrected.

Point 10: Line 518: anti-insecticidal?

Response: It was corrected.

Point 11: Line 247: A total of 54 components and 99.2 % of the total were identified. Too much of totals, revise it.

Response: According to the comment, we have rephrased the sentence.

Point 12: In Discussion section: insert links to your obtained results (Tables and Figures).

Response: The part of discussion was rewrite little be by comments other reviewers.

Round 2

Reviewer 1 Report

Comments and Suggestions for Authors

Thank you for your revisions. The manuscript has been significantly improved.